# Characterization and Immunogenicity of Influenza H7N9 Vaccine Antigens Produced Using a Serum-Free Suspension MDCK Cell-Based Platform

**DOI:** 10.3390/v14091937

**Published:** 2022-08-31

**Authors:** Min-Yuan Chia, Chun-Yang Lin, Po-Ling Chen, Chia-Chun Lai, Tsai-Chuan Weng, Wang-Chou Sung, Alan Yung-Chih Hu, Min-Shi Lee

**Affiliations:** 1Department of Veterinary Medicine, National Chung Hsing University, Taichung 40227, Taiwan; 2National Institute of Infectious Diseases and Vaccinology, National Health Research Institutes, Zhunan 35053, Taiwan

**Keywords:** influenza H7N9 vaccine, suspension MDCK cell, immunogenicity

## Abstract

Human infections with avian-origin H7N9 influenza A viruses were first reported in China, and an approximately 38% human mortality rate was described across six waves from February 2013 to September 2018. Vaccination is one of the most cost-effective ways to reduce morbidity and mortality during influenza epidemics and pandemics. Egg-based platforms for the production of influenza vaccines are labor-intensive and unable to meet the surging demand during pandemics. Therefore, cell culture-based technology is becoming the alternative strategy for producing influenza vaccines. The current influenza H7N9 vaccine virus (NIBRG-268), a reassortant virus from A/Anhui/1/2013 (H7N9) and egg-adapted A/PR/8/34 (H1N1) viruses, could grow efficiently in embryonated eggs but not mammalian cells. Moreover, a freezing-dry formulation of influenza H7N9 vaccines with long-term stability will be desirable for pandemic preparedness, as the occurrence of influenza H7N9 pandemics is not predictable. In this study, we adapted a serum-free anchorage-independent suspension Madin-Darby Canine Kidney (MDCK) cell line for producing influenza H7N9 vaccines and compared the biochemical characteristics and immunogenicity of three influenza H7N9 vaccine antigens produced using the suspension MDCK cell-based platform without freeze-drying (S-WO-H7N9), the suspension MDCK cell-based platform with freeze-drying (S-W-H7N9) or the egg-based platform with freeze-drying (E-W-H7N9). We demonstrated these three vaccine antigens have comparable biochemical characteristics. In addition, these three vaccine antigens induced robust and comparable neutralizing antibody (NT; geometric mean between 1016 and 4064) and hemagglutinin-inhibition antibody (HI; geometric mean between 640 and 1613) titers in mice. In conclusion, the serum-free suspension MDCK cell-derived freeze-dried influenza H7N9 vaccine is highly immunogenic in mice, and clinical development is warranted.

## 1. Introduction

Low pathogenic avian-origin influenza A/H7N9 viruses emerged in March 2013, and they are novel reassortants bearing H7 and N9 genes from wild bird viruses and internal genes from domestic poultry H9N2 viruses in China [1,2]. Until now, China had six waves of H7N9 outbreaks, and the highest number of human infections (766 cases) was recorded during the fifth wave (1 October 2016 to 30 September 2017) [3]. The Food and Agriculture Organization demonstrated that the high pathogenic H7N9 was established in a live poultry market in the fifth wave [4]. From 2013 to 2018, 1568 laboratory-confirmed H7N9 human cases with a case fatality rate of 39.2% were reported in mainland China. Influenza H7N9 viruses only cause sporadic zoonotic transmission at the moment but could potentially evolve to obtain the capability of efficient human-to-human transmission via reassortments or mutations [5]. Although antiviral drugs for treating influenza are important public countermeasures, vaccination is the cornerstone for preventing morbidity and mortality during pandemics.

The cell-based vaccine production platform is an attractive alternative to the egg-based platform due to its reliable production efficiency, including well-established logistics and fast scale-up [6,7]. MDCK cells are the major cell line which could efficiently grow influenza viruses to high titers [8,9]. Traditionally, these anchorage-dependent MDCK cells need to attach on roller battles or microcarriers, which is often challenging for manufacturers due to the labor-intensive process and the high cost of microcarriers. The use of suspension cells would greatly facilitate the proliferative process by eliminating the burdens of cell reattachment and medium refreshment steps [10,11]. As far as we know, two suspension MDCK cell lines, 33016PF and MDCK-SKY3851, were adapted from adherent MDCK cells (CCL34) by the serial passaging approach in a proprietary medium and used for the commercial production of seasonal influenza vaccines [12,13,14]. However, the MDCK cell-derived influenza vaccines are not widely available due to higher production costs [15]. Therefore, it is desirable to reduce the production cost of MDCK cell-derived influenza vaccines through process and formulation developments.

Since it is not predictable when influenza H7N9 viruses will obtain the capability of efficient human-to-human transmission, the stockpiling and deployment of influenza H7N9 vaccines are critical strategies for pandemic preparedness. Compared to liquid formulation, freeze-dried influenza vaccines could potentially increase storage stability and distribution flexibility, especially avoiding cold-chain requirements in developing countries [16]. Influenza subunit and split vaccines have shown well-preserved antigenicity and immunogenicity after the freeze-dried process [17,18]. However, the lyophilization of the whole inactivated influenza virus is more sensitive to the freezing and drying stress [19]. For maintaining the quality of vaccine conservation, the biochemical characteristics and immunogenicity of the freeze-dried H7N9 vaccine antigen should be evaluated. Aluminium hydroxide is a stable and safe adjuvant that is widely used in commercial vaccines and is also tested in the present study [20].

Previously, we had successfully developed a microcarrier-based adherent MDCK cell culture system for producing H5N1 and H7N9 influenza vaccines using flow-through chromatography [21,22]. All of the above bulks met the regulatory requirements of vaccines for human use and obtained 33–46% hemagglutinin (HA) recovery yields. Meanwhile, the H7N9 vaccine can elicit protective immunity against A/Anhui/1/2013 (H7N9) influenza virus infection in ferrets and induce desirable antibody responses in healthy adults [22,23]. Recently, we developed a suspension MDCK cell line for producing a liquid formulation of influenza H7N9 vaccines [24,25]. In the present work, we further evaluated the biochemical characteristics and immunogenicity of freeze-dried influenza H7N9 vaccine antigens produced using suspension MDCK cells.

## 2. Materials and Methods

### 2.1. Cell Line and Preparation of Vaccine Antigens

The serum-free suspension MDCK cell line was originally derived from adherent MDCK cells (ATCC, CCL-34) through serial adaptation processes [25]. The proprietary suspension MDCK cells were cultivated in the chemically defined BalanCD^®^ Simple MDCK medium (FUJIFILM Irvine Scientific, Santa Ana, CA, USA) supplemented with 4mM glutamine, and the suspension adaptation process was previously shown in detail (PCT Patent No. WO2017072744A1). MDCK cells were seeded at 5 × 10^5^ cell/mL in 125 mL spinner flasks (Corning Incorporated, Somerville, MA, USA). The spinner flasks were placed on a 50 rpm stir plate in a 37 °C incubator with 5% CO_2_.

Standard H7N9 influenza vaccine virus [NIBRG-268; A/Anhui/1/2013 (H7N9); egg-based lived virus] and vaccine antigen (egg-based inactivated NIBRG-268 virus with freeze-drying; NIBSC code: 14/250; E-W-H7N9) were generated using reverse genetics [HA and NA genes of A/Anhui/1/2013 (H7N9), six internal genes of A/PR/8/1934 (H1N1)] and imported from the National Institute for Biological Standards and Control (NIBSC), UK. The egg-derived influenza H7N9 vaccine virus was adapted to MDCK cells, as previously described [21]. In the present study, when the suspension MDCK cell density reached a proper range (1.8 × 10^6^ cells/mL), the 10^−4^ MOI of the H7N9 vaccine virus (NIBRG-268) was infected with the supplement of tosyl phenylalanyl chloromethyl ketone (TPCK)-trypsin (Sigma-Aldrich, St. Louis, MO, USA). After 72 hours, the culture medium was harvested for further purification [21]. Briefly, the whole H7N9 viral particles were concentrated by a tangential flow filtration (TFF) unit with a 300 kDa PESU membrane cassette (Sartorius, Göttingen, Germany) and then separated using Capto Q and Capto core 700 anion exchange chromatography columns (GE Healthcare, Chicago, IL, USA). Purified vaccine antigens were inactivated by formaldehyde and further prepared as liquid formulation (100 μg/mL in PBS; S-WO-H7N9) or freeze-drying formulation (100 μg/mL in PBS with 1% sucrose; S-W-H7N9). In addition, the viral RNA was extracted and amplified using the one-step RT-PCR (Promega, Madison, WA, USA), and the nucleotide sequences of two viral surface glycoproteins, HA and neuraminidase (NA), were determined using the ABI 3730 XL DNA Analyzer (Applied Biosystem Inc., Bedford, MA, USA).

### 2.2. Freeze-Drying

Freeze-drying is a sublimation method that removes ice crystals from frozen material. Briefly, 1 mL aliquots of the formulated vaccine containing 1% sucrose were loaded in a 2 mL cryovial (Corning Incorporated) and frozen in liquid nitrogen. The vial trays were placed in the lyophilizer vessel, which had been pre-cooled to −40 °C and held at this temperature. A dry scroll pump, connected with the lyophilizer, was used to vacuum the chamber for 24 h. When the freeze-drying was finished, the vacuum pressure was gently released, and the freeze-dried vaccine was stored at −80 °C for 1 month before subsequent biochemical and animal experiments.

### 2.3. Electron Microscopy and Nanoparticle Size Assays

In the electron microscopy assay, a carbon-coated 200 mesh copper grid was loaded with different influenza vaccine antigens for 1 min at room temperature, and the extra buffer was carefully removed from the grid using filter paper [26]. Afterwards, the grid was stained with 2% uranyl acetate solution for 3 min, dried for 12 h at room temperature and observed on a Bio-Transmission Electron Microscope (Hitachi HT7700).

In the nanoparticle size analysis, all the samples were serially diluted with PBS to stay within the operating range for analysis, and the size distribution was measured using the dynamic light scattering method (90Plus Particle Size Analyzer; Brookhaven Instruments Co., New York, NY, USA) based on the manufacturer’s instructions.

### 2.4. Single Radial Immunodiffusion Assay

The HA protein concentrations of the H7N9 influenza viruses in either liquid or freeze-dried formulations were quantified using the single radial immunodiffusion assay (SRD) [27]. The S-WO-H7N9, S-W-H7N9 and E-W-H7N9 (NIBRG-268 vaccine antigen as the reference standard antigen) were treated with 1% of Zwittergent 3-14 (Sigma, St. Louis, MO, USA) at room temperature for at least 30 min. The standard antigen was prepared at four concentrations (37, 25, 18.5 and 9.25 μg/mL) for the standard curve. Serial dilution was performed in treated samples at four concentrations to fit the linear range of the standard. For preparing diffusion gels, the reference anti-serum (anti-A/Anhui/1/2013 HA serum; NIBSC code: 13/180) was added to 1% agarose dissolved in phosphate-buffered saline (PBS) containing 0.01% Sodium azide and filled in a plastic backing. After the agarose solidified, the wells were punched and filled with 20 μL of each diluted standard antigen and test sample. The antigen–antibody reaction diffused at room temperature in a humidified chamber for 20 h and caused a zone of precipitation around the well. When the reaction was completed, the gel was washed, dried, stained by Coomassie Blue (Merck, Kenilworth, NJ, USA) and destained. The precipitation zones were recorded and measured for the determination of HA content based on the reference standards. All samples were measured in duplicate.

### 2.5. Hemagglutination (HA) Assay

All the HA assays have followed the guidelines of the World Health Organization. Two-fold serial dilutions of samples from different vaccine antigens were mixed with 0.5% turkey red blood cells (TKRBCs) in 96-well v-bottom microtiter plates (Thermo, Waltham, MA, USA) at room temperature for 30 min, and then the plates were tilted for the inspection of the HA unit (HAU).

### 2.6. SDS-Page and Western Blot Assays of Deglycosylated Vaccine Antigens

The gel electrophoresis of proteins was performed using the NuPAGE Bis-Tris precast gel system (Invitrogen, Waltham, MA, USA). The amount of protein in each sample was adjusted as required according to the SRD assay, as described above. The viral proteins were deglycosylated using PNGase F (New England Biolabs, Ipswich, MA, USA), according to manufacturer’s instructions. Briefly, the viral concentrates were denatured in the glycoprotein buffer (0.5% SDS, 40 mM DTT) at 95 °C for 5 min and were incubated with a 1/10 dilution of PNGase F in the reaction buffer at 37 °C overnight. The samples were mixed with a 1/6 dilution of loading dye (with 2% β-mercaptoethanol as the reducing agent), heated to 95 °C for 5 min, loaded onto the 4–12% Bis-Tris gel (Invitrogen) and run in the MOPS buffer (Invitrogen) at a constant voltage of 120 V for 90 min. The gel was stained in Colloidal Blue stain (Invitrogen) and destained in reaction buffer (10% methanol, 7% glacial acetic acid) until clear protein bands were revealed [28].

For the Western blot analysis, the proteins were transferred to a nitrocellulose membrane, which was then probed with a primary antibody (anti-A/Anhui/1/2013 HA serum; NIBSC code: 15/248) and a peroxidase-labeled secondary antibody. The blots were analyzed by the TMB substrate solution (Thermo).

### 2.7. Mouse Immunization Studies

Thirty-six 4-to 6-week-old healthy female BALB/c mice (BioLASCO Taiwan Co., Taipei, Taiwan) were known as specific pathogen-free and were confirmed to be H7N9, H3N2 and H1N1 seronegative by an HI assay. Mice from different groups were individually housed in plastic cages, and six treatment groups (six mice per group) containing different vaccine antigens were compared, including PBS (negative control), 0.2 μg S-WO-H7N9, 0.05 μg S-W-H7N9, 0.2 μg S-W-H7N9, 0.05 μg E-W-H7N9 and 0.2 μg E-W-H7N9. The vaccine antigens composed of inactivated influenza viruses and 300 μg aluminium hydroxide (Sigma, St. Louis, MO, USA) were homogenized by a rotating roller for 60 min. The mice were immunized intramuscularly at the quadriceps with three doses of the vaccine antigen at a 2-week interval, and sera were collected on 0, 14, 28 and 42 days post-immunization (DPI) for measuring HI and neutralizing antibody titers.

### 2.8. Hemagglutinin Inhibition (HI) Assay

The HI titers of the immunized mice sera were determined as previously described [29]. Briefly, the serum samples were incubated with receptor-destroying enzymes (Sigma) overnight and inactivated at 56 °C for 30 min. The sera were two-fold serial dilutions with PBS in 96-well v-bottom microtiter plates and were mixed with the 4 HA unit of the vaccine antigen for 15 min. The TKRBCs were added, and the highest dilution of the serum-inhibiting hemagglutination was recorded as the HI titer. A reference control serum with an acceptable range was included to confirm the assay accuracy.

### 2.9. Neutralizing Antibody (NT) Assay

The NT in the sera against the H7N9 virus was measured using the microneutralization assay, as described previously [22]. The samples of the serum were inactivated at 56 °C for 30 min to abolish the serum complement and were serially diluted in DMEM. The diluted sera were mixed with 100 TCID_50_ of the NIBRG-268 reference virus in 96-well microtitration plates at 37 °C for 2 h. The mixture was then transferred to 96-well flat-bottom plates containing confluent adherent MDCK cells (CCL34). After incubation at 37 °C with 5% CO_2_ for 4 days, the highest dilution of serum that blocked the cytopathic effect was considered as the NT titer.

### 2.10. Statistical Analysis

The data were analyzed by the SPSS20 for Windows statistic software (IBM). The Mann–Whitney–Wilcoxon test was performed for the comparison of means between two independent groups. A *p*-value of less than 0.05 was considered significant.

### 2.11. Ethics Statement

All the animal experiments were conducted in the Laboratory Animal Center of National Health Research Institutes with approval (permit number: NHRI-IACUC-104075-A), following the Institutional Animal Care Committee Guidebook published by the US Office of Laboratory Animal Welfare.

## 3. Results

### 3.1. Biochemical Characteristics of H7N9 Vaccine Antigens

To evaluate the size distribution of the vaccine antigens, the three H7N9 vaccine antigens (S-WO-H7N9, S-W-H7N9 and E-W-H7N9) were examined with laser light scattering particle size analysis. As shown in Figure 1, the S-WO-H7N9 and E-W-H7N9 vaccine antigens were composed of homogenous particles with a unimodal distribution, with an average size of 102 nm and 133 nm, respectively. Interestingly, the S-W-H7N9 vaccine antigens contained two different sizes (a major peak in 149.7 nm and a minor peak in 1000 nm) with a bimodal distribution. These results indicate that the sizes of the egg and the MDCK cell-derived viral particles may slightly increase during freeze-drying. After several passages, the suspension MDCK-based H7N9 vaccine antigens showed HA and NA nucleotide sequences identical with the original H7N9 virus grown in adherent MDCK cells, which indicates the genetic stability of the suspension MDCK cell-derived influenza H7N9 vaccine viruses (data not shown).

The morphology of the H7N9 vaccine antigens was observed using transmission electron microscopy (Figure 2). All three influenza H7N9 vaccine antigens have similar spherical particles, and the diameters of the three particles were in agreement with the particle size analysis.

These three vaccine antigens were further characterized using SDS-PAGE and Western blot assays. In the SDS-PAGE assay, HA1 and HA2 proteins are highly glycosylated, so they could not form a clear band before deglycosylation (around 48 and 30 kDa). After deglycosylation, the HA1 and HA2 proteins could form a clear band at around 36 and 22 kDa (Figure 3A) in the SDS-PAGE assay. The Western blot assay further confirmed the location of the HA1 and HA2 proteins before and after deglycosylation (Figure 3B). Overall, these three vaccine antigens have similar profiles in the SDS-PAGE and Western blot assays.

### 3.2. Quality Control of Vaccine Antigens

Since the influenza HA protein is the major antigen for inducing neutralizing antibody responses, the quantification of the HA protein with correct structural conformation is critical for the quality control of protein-based influenza vaccines. To verify the impact of lyophilization on the HA structure, the biological activity of the H7N9 vaccine antigens was assessed using a hemagglutination assay and SRD, which is the in vitro potency assay recommended by the World Health Organization. As shown in Table 1, the HA activity and concentration of S-WO-H7N9 (8192 HAU/50 μL; 108.9 μg/mL) were comparable with those of S-W-H7N9 (8192 HAU/50 μL; 106.1 μg/mL), which indicates that the freeze-drying procedures did not change the conformation structure of HA proteins in influenza H7N9 virus particles.

### 3.3. Immunogenicity in Mice

We compared the immunogenicity of three vaccine antigens (S-WO-H7N9, S-W-H7N9 and E-W-H7N9) in mice. After three vaccine doses, the serum samples from mice were evaluated for the presence of H7N9-specific HI and NT titers. After the first and second vaccination, all of the experimental and control mice did not show any obvious clinical symptoms, indicating that the number of antigens and the amount of adjuvant were safe for mice. As shown in Figure 4, all vaccine groups (except the negative control) developed robust NT and HI titers at 28 (14 days post-dose-2) and 42 (14 days post-dose-3) days post-immunization, and the NT titers were usually higher than the HI titers. At day 42 DPI, the high-dose (0.2 μg) groups of mice immunized with S-W-H7N9 and E-W-H7N9 elicited slightly higher HI and NT titers than those in the low-dose (0.05 μg) groups, although the differences were not statistically significant. At day 28 DPI, there was no statistically significant difference between the high-dose groups and low-dose groups in the HI titers of mice immunized with S-W-H7N9 and E-W-H7N9. However, the high-dose group of mice immunized with S-W-H7N9 elicited significant higher NT titers (*p* = 0.032) than the low-dose group. In addition, the mice immunized with S-WO-H7N9 or S-W-H7N9 elicited HI and NT titers comparable with those of the mice immunized with E-W-H7N9. These results indicated that the suspension MDCK cell-based H7N9 vaccine antigen can induce robust immune responses, and the freeze-drying process did not reduce the immunogenicity of H7N9 vaccine antigens in mice.

## 4. Discussion

Influenza H7N9 viruses with pandemic potentials have been circulating in China since 2013 and pose dual threats to human health and the poultry industry. Hence, the development of influenza H7N9 vaccines is desirable for pandemic preparedness. During the 2009 H1N1 pandemic, the egg-based platform could not meet the surging demand for influenza H1N1 vaccines in the first 6 months of the pandemic [30]. Therefore, alternative production systems such as insect and mammalian cell-based platforms become attractive [31]. MDCK cells expressing both sialic alpha 2–3 and 2–6 receptors have been widely used to manufacture influenza vaccines due to the high yields of viruses [32]. Basically, traditional MDCK cells require the attachment to the surface of culture flasks or microcarriers for proliferation, and the suspension culture of traditional MDCK cells is known to induce extensive cellular aggregation and apoptosis [33]. Therefore, several methods had been described to stably grow suspension MDCK cells, including the adaption of MDCK cells in a serum-free medium, in a medium containing metalloendopeptidase and in a continuous stirred-tank bioreactor [34,35] and transfection with the human *siat7e* gene (regulation of cellular adhesion) in the MDCK cell [36].

In our previous studies, we successfully produced the liquid formulation of influenza H7N9 vaccines using adherent and suspension MDCK cells [21,24,25]. Since it is not possible to predict when influenza H7N9 viruses will cause a pandemic, it is desirable to develop a freeze-drying formulation of influenza H7N9 vaccines with long-term stability for stockpiles. In the present study, we compared the biochemical characteristics and immunogenicity of three influenza H7N9 vaccine antigens produced using suspension MDCK cells (S-WO-H7N9 and S-W-H7N9) or egg-based platforms (E-WO-H7N9). The HA protein of influenza viruses is the major antigen for inducing neutralizing antibody responses and needs to preserve the critical conformation structure to maintain potency during long-term storage [37]. Freeze-drying involves the removal of water or other solvent from frozen materials by sublimation under reduced pressure [38]. During the process, changes to the protein structure might lead to a loss of vaccine potency. Therefore, a stabilizer, such as sucrose (non-reducing sugar), should be included in the formulation to increase the stability of vaccine antigens in the solid phase, and the characteristics and activities of HA proteins should be evaluated before and after freeze-drying [37]. In the present study, the increase in particle size after freeze-drying could result from the mechanical changes including ice crystallization and osmolarity, which alter the virus particle structure and lipid membrane [37]. However, the freeze-drying process does not alter the biochemical characteristics of the HA protein, as monitored using SDS-PAGE and Western blot assays. In addition, we used the WHO-recommended in vitro potency assay (SRD) to specifically detect the immunologically active HA protein [39] and found that the freeze-drying process did not change the binding capability between HA proteins with functional antibodies in the sheep standard serum. For evaluating the HA potency before and after freeze-drying, the SRD and HA assays of S-WO-H7N9 and S-W-H7N9 were performed, and these two vaccine antigens had comparable biological activities of HA proteins. The results indicate that the HA potency of H7N9 vaccine antigens was not influenced by the freeze-drying treatment in this study.

The amount of glycosylation on the HA protein could vary significantly in different influenza strains and subtypes, which could apparently impact the molecular weight of the glycoproteins. In the previous study, Harvey et al. had demonstrated that the deglycosylation of H3N2 and H1N1 viruses can induce the migration of their HA1 and HA2 bands in SDS-PAGE and improve the quantitation of HA proteins [28]. Here, we also found that deglycosylation could alter the migration patterns of the HA1 and HA2 proteins in SDS-PAGE and Western blot assays. This modification could obtain a more accurate evaluation of H7N9 vaccine antigens, which is consistent with a previous study for quantifying the HA contents of influenza H7N9 vaccine antigens [40]. It is noted that the NA protein of H7N9 cannot be clearly observed in gel before or after deglycosylation. It could be speculated that the current quantity was too low to present visible bands, and the results were similar with other strains of influenza viruses [28,41].

Despite the gold standard for influenza vaccine and infection research being the ferret model, the mouse model is commonly used for immunogenic assays in the pilot study due to the availability of reagents, the plethora of genetic tools and the ease of husbandry. The criteria for the evaluation of immunogenicity in influenza vaccines are usually based on serological measures (HI or NT assays). The previous studies had demonstrated that the HI and NT titers greater or equal to 40 and 80, respectively, correlated with protection against influenza virus infection [42,43]. In the present study, S-WO-H7N9 or S-W-H7N9 vaccine antigens were found to be able to elicit robust immune responses against influenza H7N9 virus in mice and had comparable immunogenicity with the standard antigen (E-W-H7N9). Meanwhile, the results showed no significant differences in the NT and HI titers between the low- and high-dose groups throughout the study, which indicates that 0.05 μg of HA antigens had enough potential to induce immune responses and reach the plateau phases. The serum-free suspension MDCK cell culture was only performed in small spinner flasks in the present study. However, we have tried to scale up the volume of the suspension MDCK cell culture to 5L using a bioreactor system, and the H7N9 qualities between the spinner flask and bioreactor systems are similar. Briefly, The HA titers produced using the spinner flask and bioreactor systems are 8192 and 18,428 HAU/50μL, respectively. The concentrations of HA proteins produced using the spinner flask and bioreactor systems are 106.1 and 122.7 μg/mL, respectively [24].

In contrast to seasonal influenza viruses, the influenza H7N9 virus is a novel subtype, and human populations are immunologically naïve to it. In previous clinical trials, the immunogenicity of influenza H7N9 split vaccines is generally considered to be weaker than that of the seasonal influenza split vaccines [44]. However, the H7N9 whole virus vaccine candidate in our study achieved a robust immune response in the mouse model. One possible explanation for this may be related to the fact that the whole-virion vaccine was used in the present study. Compared with the influenza split and subunit vaccines, the previous studies have demonstrated that whole-virion vaccines can induce stronger cell-mediated immunity toward the Th1 immune response and stimulate monocytes to secrete IL-2, IL-12 and IFN-γ by the dendritic cells [45]. Furthermore, a previous clinical trial has demonstrated that the liquid formulation of the adherent MDCK cell-based whole-virion H7N9 vaccine adjuvanted with aluminium hydroxide can induce adequate HI and NT titers in young adults [23]. In the present study, we further confirmed that the suspension MDCK cell-derived freeze-dried influenza H7N9 vaccine is highly immunogenic in mice, and clinical developments evaluating different adjuvants are warranted.

## Figures and Tables

**Figure 1 viruses-14-01937-f001:**
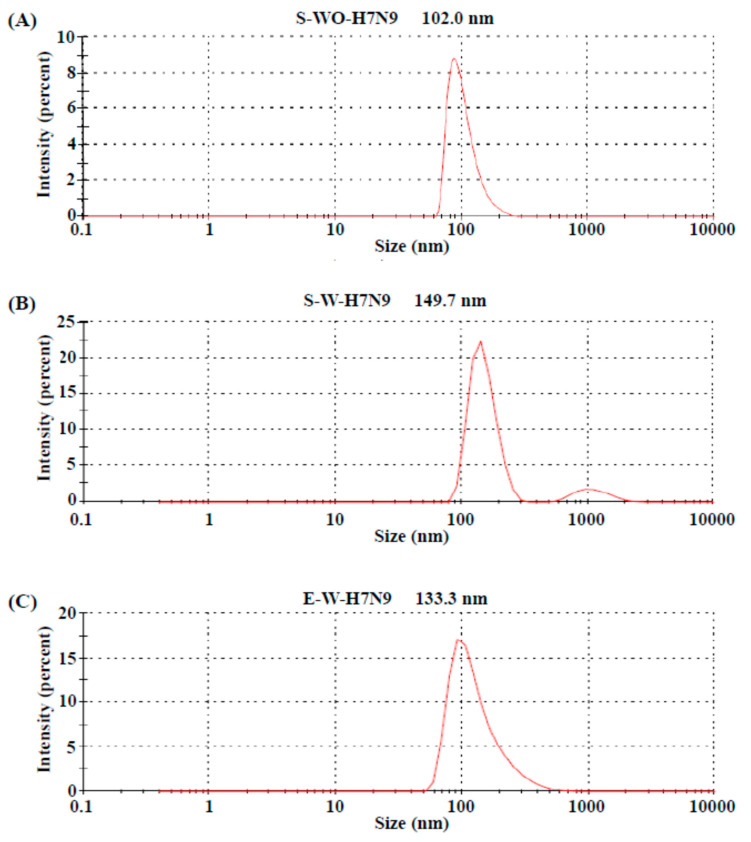
The particle size distribution of three influenza H7N9 vaccine antigens analyzed using dynamic light scattering. (**A**) S-WO-H7N9; (**B**) S-W-H7N9; (**C**) E-W-H7N9.

**Figure 2 viruses-14-01937-f002:**
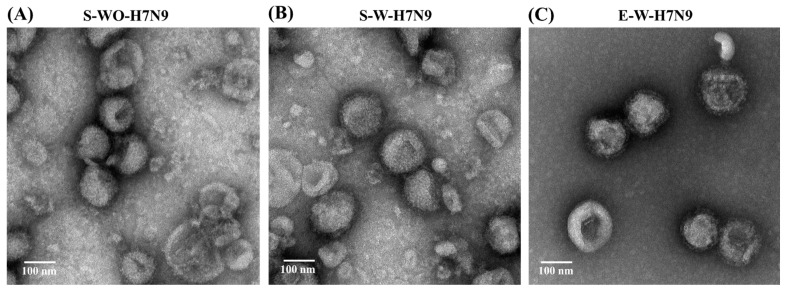
Morphology of three influenza H7N9 vaccine antigens analyzed using transmission electron microscopy. (**A**) S-WO-H7N9; (**B**) S-W-H7N9; (**C**) E-W-H7N9. Bar = 100 nm.

**Figure 3 viruses-14-01937-f003:**
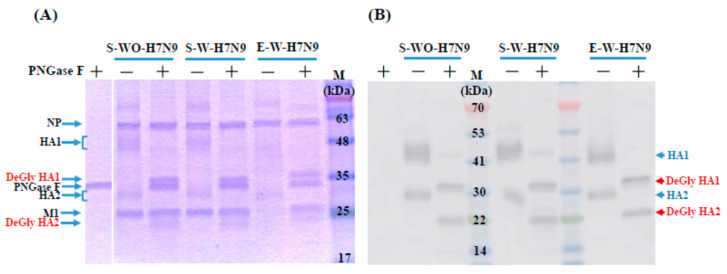
SDS-PAGE and Western blot analyses of three H7N9 vaccine antigens. (**A**) The first lane was only loaded with PNGase F, and then an equal number of antigens was loaded in the following lanes with or without deglycosylation using PNGase F. After electrophoresis, the gel was visualized by Colloidal Blue staining. (**B**) Antigens were detected by anti-A/Anhui/1/2013 HA serum (NIBSC code: 15/248) in the nitrocellulose membrane. NP: nucleoprotein; M1: matrix protein; HA: hemagglutinin; DeGly: deglycosylation; +: the line with PNGase F; −: the line without PNGase F.

**Figure 4 viruses-14-01937-f004:**
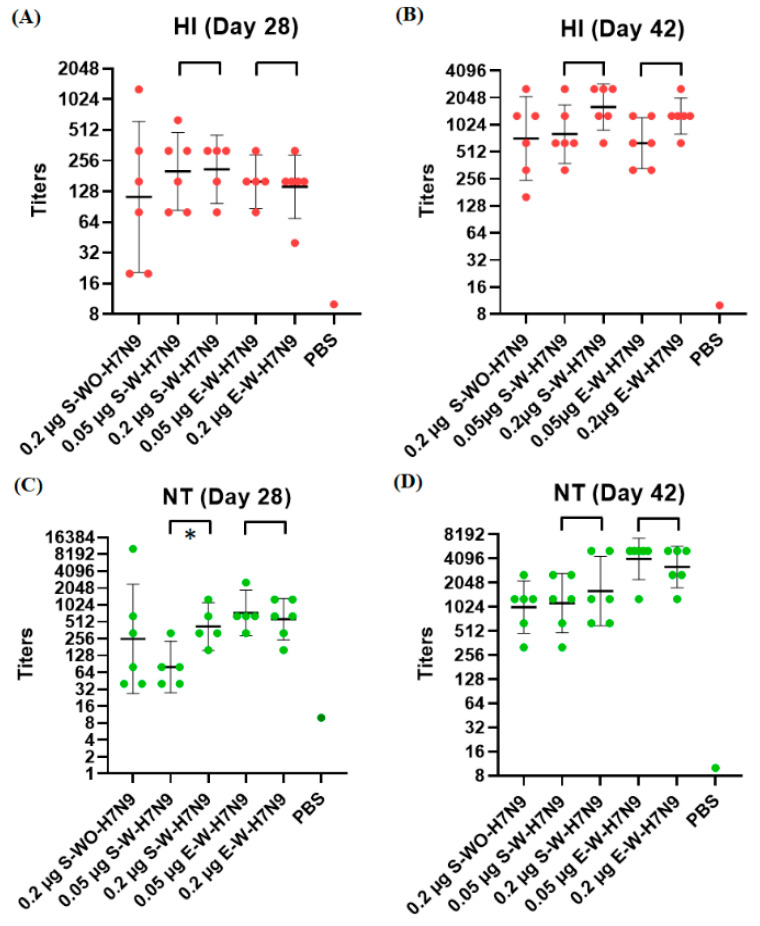
Serum HI and NT titers against H7N9 viruses in mice immunized with three H7N9 vaccine antigens. The sera were collected at 28 DPI (14 days post-dose-2) [(**A**,**C**)] and 42 DPI (14 days post-dose-3) [(**B**,**D**)] to measure the hemagglutination inhibition and neutralizing antibody titers and are expressed as the geometric mean ± 95 confidence interval. HI: hemagglutination inhibition; NT: neutralizing antibody. *: significant difference (*p* < 0.05).

**Table 1 viruses-14-01937-t001:** The results of hemagglutination activity and single radial immunodiffusion assays in different vaccine antigens.

	S-WO-H7N9	S-W-H7N9	E-W-H7N9
**HAU (50 μL)**	8192	8192	128
**SRD (** **μg/mL)**	108.9	106.1	37.0

HAU: hemagglutination activity unit; SRD: single radial immunodiffusion assays.

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
