# Peer review of "Characterization and Immunogenicity of Influenza H7N9 Vaccine Antigens Produced Using a Serum-Free Suspension MDCK Cell-Based Platform"

_viruses, 2022, doi:10.3390/v14091937_

Round 1
Reviewer 1 Report
The work by Chia et al., extends on prior work and describes the production of vaccine material from a recommended H7N9 vaccine virus strain in a suspension cell line developed by the authors. The manuscript is concise and clear, conveying useful information. The description of results and discussion are appropriate.
For Table 1, it is important that the authors state the number of replicates used for the SRID assay either in the table legend or in the methods.
The manuscript is acceptable after a few minor edits:
Line 5: there is an extra comma within an author's name.
Line 149: "suitable" is an inadequate description, I recommend stating the percentage of turkey RBCs used.
Line 154: performed using NuPAGE
Line 174: I recommend a semi-colon after "compared"
Line 178: please delete "subsequent"
Line 183: please replace "Shortly" with "Briefly"
Line 195: please clarify which MDCK cells were used, presumably CCL34 adherent cells for the NT assay.
Line 211: distribution with an average
Lines 251-252: should be microgram not microliter
Line 261: I recommend "After three vaccine doses serum samples..."
Line 266: replace DPI with days post-vaccination
Line 283: "since 2013 and pose dual threats..."
Line 307: it appears some text is missing, perhaps "during the process changes to protein structure might lead..."
Line 313: delete "etc"
Reviewer 2 Report
The goal of this study was to assess the biochemical properties and immunogenicity of a serum-free suspension MDCK cell-based H7N9 influenza A vaccine following freeze-drying compared to a freeze-dried egg-based version of the vaccine. The authors use TEM, SDS-PAGE, and Western blot assays to convincingly establish that the freeze-drying process did not alter the morphology nor biochemical properties of the vaccine antigens. Further, the authors use hemagglutination and single radial immunodiffusion assays to convincingly demonstrate that the freeze-drying process did not adversely impact the structural conformation of the HA proteins present in the MDCK cell-based H7N9 vaccine. Finally, the authors use a well-established mouse immunization model to convincingly show that the freeze-drying process did not reduce the ability of the MDCK cell-based H7N9 vaccine to elicit high titers of neutralizing antibodies. The use of a freeze-dried egg-based H7N9 vaccine throughout the study was a suitable control. My comments and suggestions:
1. The authors do a good job of promoting the benefits of freeze-drying (e.g., increased storage stability, reduced cold-chain requirements). However, it is unclear from the methods and figure legends how long their freeze-dried vaccine was in storage at -80C before use in studies. Was it days, weeks, or months? Were there differences in storage times for the samples used in biochemical and animal studies? This key information needs to be added to the paper.
2. Likewise, have the authors conducted experiments (or are experiments planned) to examine the stability of their MDCK cell-based vaccine over time and/or at lower temperatures (e.g., -20C)?
3. For Figure 4 I recommend the use of a scatter plot with means (or medians) rather than bar graphs. The error bars shown indicate a range of responsiveness in mice (not unexpected). It would be informative to see whether the responses are comparably scattered between those mice given the freeze-dried and those given the non-freeze-dried vaccine. This might help establish that reconstitution after the freeze-drying process did not adversely impact the distribution of HA antigens per dose.
4. Likewise, due to the wide range of values in Figure 4, I recommend use of non-parametric test (e.g., Mann Whitney Wilcoxon test) for statistical analyses instead of the unpaired Student's t-test.
5. I recommend removal of the term "mock" from the supplement figure as it is confusing. Readers may get the incorrect impression that the lane contains lysates from uninfected MDCK cells rather than PNGase F alone.
Reviewer 3 Report
The manuscript by Chia et al., showing the Characterization and Immunogenicity of Influenza H7N9 Vaccine Antigens Produced using a Serum-Free Suspension MDCK Cell-Based Platform is overall an interesting study that tests the antigen produced by suspension MDCK and compared to egg-based platform and show that the immunogenicity is comparable. The manuscript is well written and results are clearly presented. It can be accepted for publication with moderate language changes.
Reviewer 4 Report
In their manuscript , Chia at al, describe the characterization of an influenza H7N9 vaccine produced in suspension cultures with and without freeze drying. They compared the biochemical characteristics and immunogenicity of suspension culture produced vaccines to standard egg based vaccine. They showed that suspension culture produced and freeze-dried vaccine has similar characteristics and immunogenicity compared to the standard egg based freeze-dried H7N9 vaccine.
The H7N9 influenza A virus represents a potential future pandemic virus with severe health impact. Therefore, it is very important to have efficient vaccines available. This manuscript describes the successful production and of a vaccine in cell culture and the immunogenic activity of a potential future vaccine and its long-term storage. As such, the manuscript represents a major contribution to the field.
The manuscript is well written, the results clearly presented and the discussion is appropriate. I only have some minor comments.
The suspension culture was only done in very small spinner flasks. Will similar yields and quality of material be produced if these volumes are scaled up?
Page 3, line 98: please specify exactly the virus which was used here, e.g. which segments are coming from which virus? Please provide a reference.
Page 3, line 103: please specify exactly which virus was used.
Page 3, 105: please describe briefly how the purification was performed. There is some confusion with the comment in the discussion saying that whole virion vaccine was used. Therefore, it is not clear weather purified HA and NA were used as vaccines in this study or whole virions.
Figure 3: describe in the figure legend what the plus and minus signs mean.
Figure 4: the numbers are very difficult to read. Maybe increase size. There is a difference between the low dose S-W-H7N9 and the low dose E-W-H7N9 which may be worth mentioning, even though it may not be significant. Furthermore, please state weather any differences between groups in part A or in part B were significant. If this was not the case, it should be mentioned in the legend.
The study has some limitations. See my above comment on larger volumes. Also, the gold standard for neutralizing antibody production is the ferret. The ultimate test for performance of the vaccine would be human studies. I suggest adding a brief paragraph in the discussion to mention those limitations.
